# Structural basis for inhibition of an archaeal CRISPR–Cas type I-D large subunit by an anti-CRISPR protein

M. Cemre Manav [1,3], Lan B. Van[1], Jinzhong Lin [2], Anders Fuglsang [2], Xu Peng [2✉] & Ditlev E. Brodersen [1✉]

A hallmark of type I CRISPR–Cas systems is the presence of Cas3, which contains both the nuclease and helicase activities required for DNA cleavage during interference. In subtype I-D systems, however, the histidine-aspartate (HD) nuclease domain is encoded as part of a Cas10-like large effector complex subunit and the helicase activity in a separate Cas3′ subunit, but the functional and mechanistic consequences of this organisation are not currently understood. Here we show that the *Sulfolobus islandicus* type I-D Cas10d large subunit exhibits an unusual domain architecture consisting of a Cas3-like HD nuclease domain fused to a degenerate polymerase fold and a C-terminal domain structurally similar to Cas11. Crystal structures of Cas10d both in isolation and bound to *S. islandicus* rod-shaped virus 3 AcrID1 reveal that the anti-CRISPR protein sequesters the large subunit in a non-functional state unable to form a cleavage-competent effector complex. The architecture of Cas10d suggests that the type I-D effector complex is similar to those found in type III CRISPR–Cas systems and that this feature is specifically exploited by phages for anti-CRISPR defence.

[1] Department of Molecular Biology and Genetics, Aarhus University, Gustav Wieds Vej 10c, DK-8000 Aarhus C, Denmark. [2] Department of Biology, University of Copenhagen, Ole Maaløes Vej 5, DK-2200 København N, Denmark. [3] Present address: MRC Laboratory of Molecular Biology, Cambridge CB2 0QH, UK. ✉email: peng@bio.ku.dk; deb@mbg.au.dk

CRISPR–Cas (clustered regularly interspaced short palindromic repeats–CRISPR-associated proteins)[1] systems are present in most bacteria and archaea[2] and provide sequence-specific protection against invading DNA or RNA[3–9]. CRISPR–Cas immunity is obtained through a three-step process[10,11] consisting of adaptation (also known as spacer acquisition), involving incorporation of fragments of invading DNA or RNA into genomic CRISPR arrays; expression, transcription and maturation of CRISPR-RNAs (crRNAs) carrying a sequence complementary to the invading nucleic acids and assembly of the Cas–crRNA surveillance complex; and finally, interference, cleavage of foreign nucleic acids that have been encountered before by a Cas–crRNA effector complex[3,11]. CRISPR–Cas systems have been divided into two main classes depending on whether their effector complexes consist of multiple (Class 1) or a single (Cas9, Class 2) protein, six types (I–VI) based on the configuration of the signature Cas proteins, and finally 33 subtypes on the basis of functional and mechanistic features[12]. Type I (Class 1) CRISPR–Cas systems encode a large multi-subunit effector complex termed Cascade (CRISPR-associated complex for antiviral defence), which also includes a crRNA that recognises the invading DNA sequence during the interference stage. Formation of a DNA–RNA heteroduplex (an R-loop) inside the effector complex then triggers recruitment of stand-alone helicase and nuclease activities, which unwind and cleave DNA, respectively. In most type I systems, both these functions are carried out by the signature protein Cas3[2], which contains an N-terminal metal-dependent histidine-aspartate (HD) nuclease domain, responsible for endo-nucleolytic cleavage of the target, fused to an ATP-dependent and single-stranded DNA (ssDNA)-stimulated superfamily-2 (SF2) helicase (Supplementary Fig. 1a)[13–16]. However, in type I-A and I-D systems, the activities of Cas3 are carried out by separate proteins called Cas3' (helicase) and Cas3" (nuclease). Type III CRISPR–Cas systems also have multi-protein effector complexes

for which the large subunits are members of the Cas10 family[2]. Cas10 proteins contain an N-terminal HD nuclease domain that is evolutionarily distinct from Cas3", two RNA recognition motif (RRM) domains homologous to those found in polymerases and cyclases, and a helical Thumb/C-terminal domain (Supplementary Fig. 1b)[17]. Consequently, type III effector complexes are intrinsic nucleases and do not need to recruit additional proteins for activity.

Type I-D CRISPR–Cas systems, which are found both in bacteria and archaea, are exceptional in that they contain a hybrid large subunit, Cas10d, bearing resemblance to both the signature proteins found in type I (Cas3) and type III (Cas10)[18]. Cas10d thus contains an N-terminal HD domain similar to those in type I Cas3 and Cas3" proteins, which is distinguished by lacking a circular permutation found in type III large subunit HD nucleases that results in a conserved His residue shifting from the N to the C terminal end of the domain[2]. The Cas10d HD domain is fused to a predicted inactivated polymerase consisting of two RRM domains reminiscent of those found in type III Cas10 subunits[19]. Recently, it was shown that an archaeal type I-D system from *Sulfolobus islandicus* is capable of cleaving both double-stranded DNA (dsDNA) and, remarkably, also ssDNA, which could be targeting rudivirus (SIRV) replicative intermediates[20]. dsDNA cleavage is catalysed by the Cas10d HD domain and in the presence of the Cas3' helicase and ATP, multiple cleavage events take place on both DNA strands. ssDNA cleavage, on the other hand, appears to follow a 6-nucleotide (nt) ruler-like mechanism and is catalysed by the backbone subunit Csc2 (Cas7), in a way reminiscent of the protospacer RNA transcript cleavage observed in type III systems[20]. Here, we present the crystal structure of the Cas10d large effector complex subunit from the *Sulfolobus islandicus* I-D system, both in isolation and in complex with the *Sulfolobus islandicus* rod-shaped virus 3 (SIRV3) anti-CRISPR protein, AcrID1. The structures reveal an unusual domain architecture of the large subunit, which supports that the effector

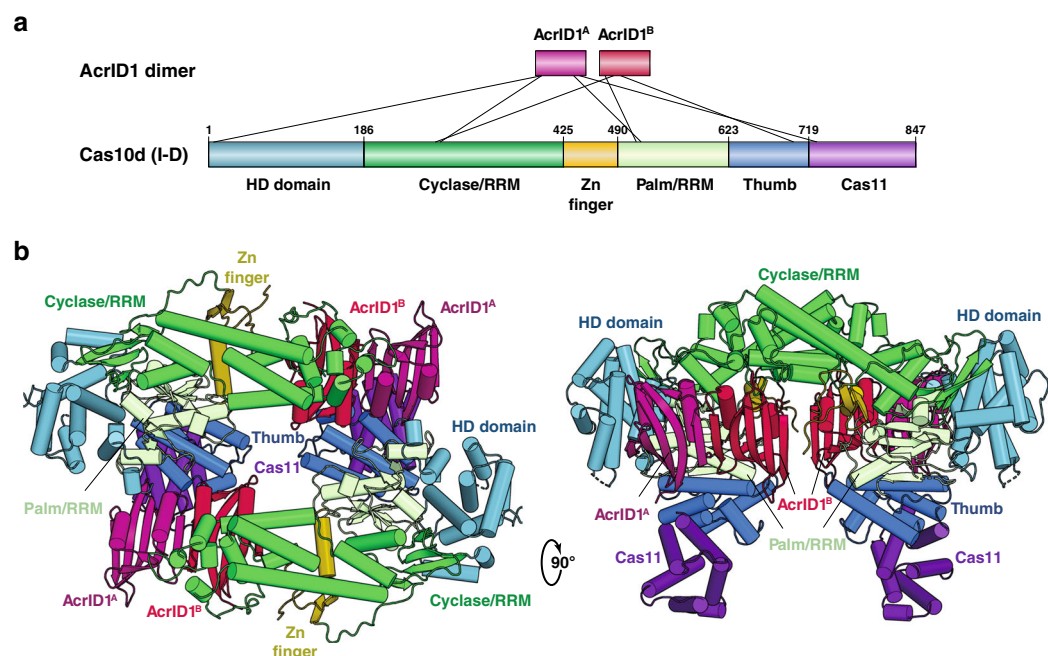

**Fig. 1 Overview of the *S. islandicus* Cas10d-SIRV3 AcrID1 complex. a** Interaction map showing the two monomers of the SIRV3 AcrID1 dimer (A and B, purple and red, top) and the individual domains of the *S. islandicus* Cas10d protein (as indicated, bottom). Residue numbers at domain borders are indicated. **b** Overview of the 2:4 complex of Cas10d and AcrID1 in two orthogonal views with colours as in a. Left, top view with the two molecules of Cas10d in the top left and bottom right corner of the structure (blue and green), bridged by two AcrID1 dimers (red/purple); Right, side view with the two molecules of Cas10d on the left and right.

complex is similar to those found in type III CRISPR–Cas systems. Moreover, we show that binding of the anti-CRISPR protein sequesters Cas10d in a non-functional state, in which it cannot engage in interference. Taken together, our data provide a framework for understanding essential mechanistic features that distinguish type I-D from other type I and type III CRISPR–Cas subtypes.

## Results

**S. islandicus Cas10d and SIRV3 AcrID1 form a circular higher-order complex**. The crenarchaeon *Sulfolobus islandicus* LAL14/1 carries one subtype I-D CRISPR–Cas locus containing spacers matching the genome of the lytic rod-shaped virus, *Sulfolobus islandicus* rod-shaped virus 2 (SIRV2), for which it is a laboratory host[21]. To counter the CRISPR–Cas immunity, SIRV2 on its part expresses a 12.2 kDa anti-CRISPR (Acr) protein, AcrID1, which is conserved in many archaeal viruses including *Sulfolobus islandicus* rod-shaped virus 3 (SIRV3)[22]. SIRV3 AcrID1 was shown to inhibit type I-D immunity in *S. islandicus* through direct binding to the large subunit of the effector complex, Cas10d[23]. To understand the molecular architecture of the unique type I-D system as well as mechanism of inhibition by AcrID1, we reconstituted the Cas10d–AcrID1 complex in vitro and determined the crystal structure to 3.5 Å resolution by single-wavelength anomalous dispersion (SAD) phasing using Se–Met substitution (Supplementary Table 1). In the complex with Cas10d, AcrID1 forms a dimer as observed in the crystal structure of the isolated anti-CRISPR protein[23], and makes extensive contacts with all major domains of Cas10d (Fig. 1a). The net result of these interactions is that two AcrID1 dimers sequester two Cas10d subunits into a large, circular 4:2 complex (Fig. 1b). The buried surface areas between the AcrID1 dimer and the two molecules of Cas10d constitute approximately 930 and 880 Å² at the two distinct sites of interaction, and size exclusion chromatography supports that this higher-order complex represents the biologically relevant form[23].

**Cas10d has a unique domain architecture**. As predicted from comparative sequence analysis, the 847 residue (94 kDa) *S. islandicus* Cas10d consists of six domains: A HD nuclease domain (1–186), a Cyclase/RRM domain (187–425), a zinc finger domain (426–490), a Palm/RRM domain (491–623), and a C-terminal helical region consisting of a polymerase-like Thumb domain (624–719) and an extension with predicted homology to Cas11 (720–847) (Figs. 1a, 2a and Supplementary Fig. 1b)[12]. The six domains are arranged in a right-angled wedge with the Cyclase/RRM and Cas11 domains at the extremes. The overall structure of the Cas10d subunit and relative domain organisation is unique when compared to the closest type III Cas10 orthologues for which structures are available (type III-A Csm1 and type III-B Cmr2, Supplementary Fig. 2)[24,25]. Most noteworthy is the fact that the Cyclase/RRM domain forms an extended structure that protrudes away from the core of the protein making the protein significantly more open than known type III Cas10 proteins (Fig. 2a and Supplementary Fig. 2).

**Cas10d contains a hybrid HD nuclease domain**. The HD domain is a general divalent metal-dependent hydrolytic phosphodiesterase domain[26], which is configured to function as an endo and/or exo-nuclease in CRISPR–Cas interference[14,15,24,27–29]. HD nuclease domains are found in a wide range of nucleic acid enzymes and consist of a conserved core of 6–7 helices that organise the residues of the active site[16]. Outside this core motif, the structures vary considerably. The HD domain found in Cas10d appears to be a minimalised version of the core motif found in other type I systems,

such as *Thermus thermophilus* Cas3 (Cas3Tth) and *Thermobifida fusca* Cas3 (Cas3Tfu, Fig. 2b)[15,16]. Compared to Cas3, Cas10d lacks two helices (α7 and α8), but compensates with an N-terminal extension (residues 1-26), part of which forms a helix (α1, residues 14–24) that spatially overlaps with α8 in Cas3Tth despite running in the opposite direction. Interestingly, this N-terminal topology is also found in type I-A Cas3" proteins, such as *Methanocaldococcus jannaschii* Cas3 (Cas3Mja, Supplementary Fig. 3a), which are isolated nuclease subunits[30]. The Cas10d HD domain thus appears to represent a structural hybrid between Cas3" (α1–α3, residues 13–59) and Cas3 (α2–α6, residues 26–134 and α8, residues 139–157). Finally, at its C-terminal end (residues 158–186), the Cas10d HD domain contains two additional helices (α9–α10), which spatially overlap with a helix and a short beta-hairpin in full length Cas3Tth (Supplementary Figs. 3b and 4).

**The Cas10d HD domain binds a single metal ion**. The HD nuclease domain is defined by the presence of three conserved sequence motifs, I, II, and V that each contain metal-coordinating His and Asp residues. Two of these are always adjacent in the sequence of motif II and have given rise to the name of the domain (HD)[26]. Interestingly, HD nucleases are generally not active with $Mg^{2+}$, but require one or more transition metal ions ($Mn^{2+}$, $Ni^{2+}$, $Cu^{2+}$, or $Fe^{2+}$) for activity[16]. In the active site of the Cas3Tth HD domain, two His, two Asp, and a Ser residue coordinate a single divalent metal ion[16]. Other Cas3 HD domains, such as the one found in Cas3Tfu, have an extended set of residues that includes two additional histidines that come together to coordinate two ions (Fig. 2b, left)[14]. Structural and sequence alignment with Cas10d reveals that while the core HD domain active site is very similar to that found in Cas3Tth, the additional residues that coordinate the second metal ion (His115, His149, and His150 in Cas3Tfu) are not present, suggesting that this HD domain only binds a single (divalent) cation (Fig. 2b, right, and Supplementary Fig. 4). This difference likely affects catalysis and/or specificity of the Cas10d HD domain but since neither ions nor substrate are present in this structure, the final conclusion must await further analysis.

**Cas10d contains a degenerate nucleotidyl transferase fold**. Like type III Cas10 proteins, such as Csm1 and Cmr2, Cas10d contains two domains following the HD domain with ferredoxin/RRM-like folds, which we have termed the Cyclase/RRM and Palm/RRM domains based on their predicted homology to nucleotidyl transferases[31]. The first of these RRM domains (residues 187–425) maintains the interleaved βαβ-βαβ structure characteristic of an RRM domain but has two very short β-strands and appears partially degenerate (Fig. 2a, dark green and Supplementary Fig. 5a). Moreover, the helix of the last βαβ motif contains a large insertion (residues 230–407) forming a separate, mainly helical, domain of nearly 150 residues that extends away from the core of the protein (Fig. 2a, green). Between the two RRM domains, there is a zinc finger domain (residues 426–490) involving cysteines 429, 432, 478, and 481, which come together to tightly interact with a $Zn^{2+}$ ion, similarly to what was observed for *P. furiosus* Cmr2/Cas10 (type III-B, Supplementary Fig. 2)[24]. The second RRM domain (Palm/RRM, residues 491–623) is more canonical except for a small β-hairpin insertion surrounding the helix of the first βαβ RRM motif (Supplementary Fig. 5b). This is highly reminiscent of the Palm and Fingers domains of some polymerases, such as prokaryotic DNA polymerase IV (Supplementary Fig. 5c)[32]. Due to the extended conformation of the Cyclase/RRM domain, the Palm/Fingers β-sheet structures in Cas10d appear more exposed compared to other large subunit structures where they are sandwiched between α-helices[14,24,28,29].

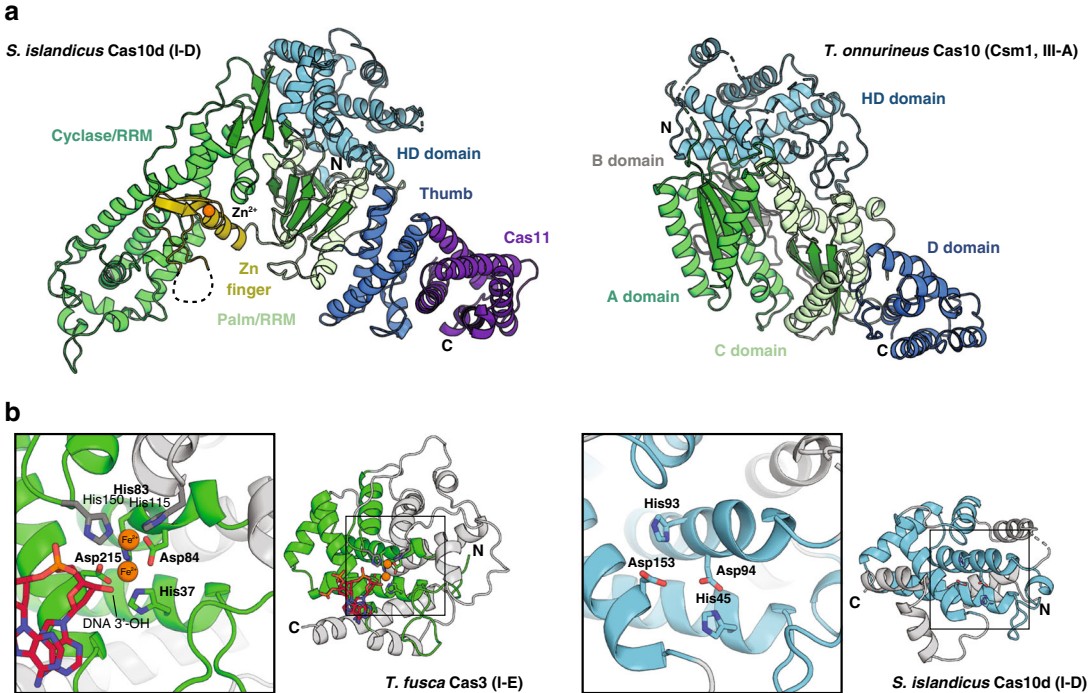

**Fig. 2 Domain structure of *S. islandicus* Cas10d. a** Left, overview of the domain structure of *S. islandicus* Cas10d (type I-D, this work) with colours as in Fig. 1. The location of the N and C termini as well as the $Zn^{2+}$ ion (orange sphere) of the zinc finger are indicated; Right, the structure of the *T. onnurineus* Cas10 (Csm1, type III-A, PDB ID 4UW2) in an approximately aligned orientation[25]. The B domain is an additional domain found adjacent to the zinc finger motif in Csm1, not present in Cas10s (Supplementary Fig. 1). **b** Details of the HD nuclease domain found in *T. fusca* Cas3 (left, type I-E, green, PDB ID 4QQW) and *S. islandicus* Cas10d (this work, right, type I-D, blue) which are structurally homologous[15]. The HD domains are aligned on the active site and the conserved core of 6–7 helices shown with colours. Highly conserved residues in the active site (HD, D, H) are shown in coloured sticks with labels in bold type in the zoomed views and additional metal-coordinating residues in *T. fusca* Cas3 with grey sticks and regular type. Two $Fe^{2+}$ ions and DNA nucleotides present in this structure are shown with orange spheres.

Interestingly, the Cas10d Palm/RRM domain is shorter than that found in many type III Cas10 proteins (e.g. Cmr2) and does not have the conserved β-hairpin motif (GGDD) nor the P-loop required for ATP binding, both of which are conserved in the Cmr2 and Csm1 type III subunits (Supplementary Fig. 6)[24,25]. Moreover, the Finger domain is placed more towards the back of the Palm domain compared to DNA polymerase IV allowing a β hairpin from the Cyclase/RRM domain to block the putative ATP binding pocket. Together, these observations suggest that the domain is not capable of nucleotide binding and thus inactive (Supplementary Fig. 5c). Conversely, the zinc finger motif is very well conserved between the large subunits of type III (Cas10 group) and Cas10d (Supplementary Figs. 1b, 2, and 6).

Finally, the C-terminal region of Cas10d consists of 10 α-helices, and is thus more similar to type III than type I large subunits[24,28], for which the domain adopts an αβ fold (Fig. 2a, dark blue and purple)[14,15,29]. The domain can be broken down into a Thumb domain similar to those found in polymerases (residues 624–719) and a helical bundle, which is structurally similar to the small effector complex subunit Csm2/Cmr5/Cas11 (residues 720–847, Fig. 2a, purple). This is consistent with comparative sequence analysis, which has suggested that several large subunits contain Cas11-like domains at their C termini[12]. Taken together, our observation that the HD nuclease domain of Cas10d is a hybrid between the type I domains found in Cas3 and Cas3" while that the rest of the protein is more similar to type III large subunits suggests that the type I-D large subunit could represent an evolutionary intermediate between canonical type I and type III CRISPR–Cas systems.

**SIRV3 AcrID1 makes extensive contacts to all major domains of Cas10d.** In the 2:4 complex of Cas10d and AcrID1 there are no direct interactions between the two Cas10d subunits but several specific interactions between both AcrID1 monomers and each domain of Cas10d that effectively cross-link the Cas10d subunits (Fig. 1b). These interactions tie one AcrID1 monomer (chain A) to the HD nuclease, Palm/RRM, and Thumb/Cas11 domains of one Cas10d subunit and the other AcrID1 monomer (chain B) to the Cyclase/RRM and C-terminal Thumb domains of another Cas10d subunit (Fig. 3). Contacts to the Cyclase/RRM domain are extensive and involve both charged and hydrogen bond interactions to two exposed loops (273–275 and 292–307, Fig. 3a). Although these regions contain some similarity to Csm1/Cmr2, direct contacts appear to be specific and uniquely targeting Cas10d (Supplementary Fig. 6). The interactions between AcrID1 and the HD nuclease domain are especially intriguing because they involve an N terminal extension of Cas10d (residues 1–13) for which specific contacts are made to both Asn3 and Arg6 (Fig. 3b). Interestingly, this extension is not part of the structured HD domain and likely another unique feature of Cas10d (Supplementary Fig. 4). Structural alignment with divalent metal-containing HD domains shows that particularly His45, which is the active site residue closest to the N-terminus, is further away from the rest of the active site than in metal-bound Cas3 HD domain structures. Consequently, this could either be a result of the missing ion or AcrID1 interaction. Contacts between AcrID1 and the Palm/RRM domain do not appear as strong as those to the Cyclase/RRM domain and most involve the exposed loop region 516–531, which is also not present in type III CRISPR–Cas large subunits (Fig. 3c and Supplementary Fig. 6). Finally, a few

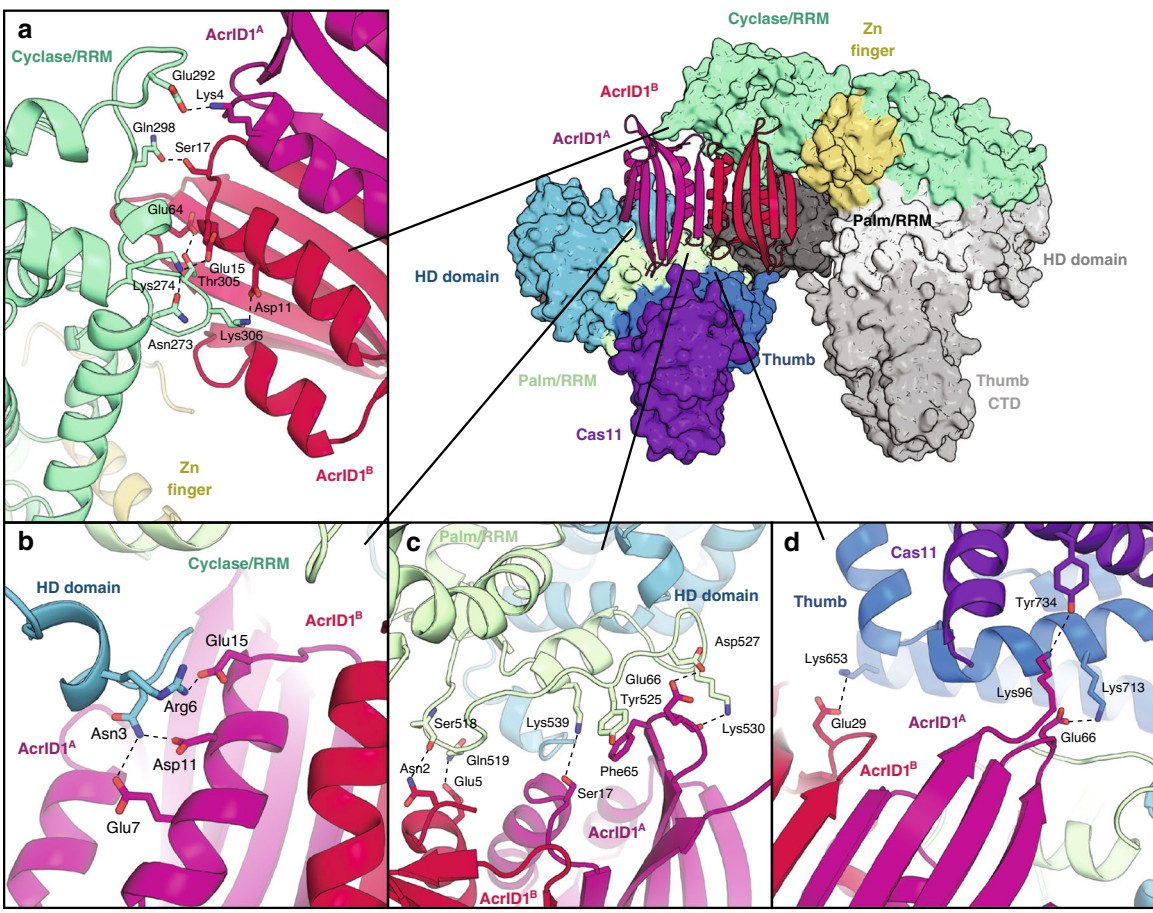

**Fig. 3 Details of the interactions between Cas10d and AcrID1.** In all panels, hydrogen bonds and charged interactions are shown with labelled residues, dashed lines and with colours as in Fig. 1. **a** Interactions between AcrID1[A] and the Cyclase/RRM domain with relevant residues shown as labelled sticks. **b** Interactions between AcrID1[A] and the HD domain. **c** Interactions between both monomers of AcrID1 and the Palm/RRM domain. **d** Interactions between both monomers of AcrID1 and the Thumb/Cas11 domains.

contacts exist between AcrID1 and the Thumb/Cas11 domains of Cas10d (Fig. 3d).

**AcrID1 prevents formation of a cleavage-competent I-D effector complex on target DNA.** To understand the functional consequences of AcrID1 binding, we analysed the effect of AcrID1 on dsDNA cleavage by a complete type I-D effector complex reconstituted in vitro[20]. I-D backbone complex consisting of Csc1 (Cas5), Csc2 (Cas7) and crRNA was mixed with Cas10d co-purified with SS (small subunit, Cas11), Cas3', ATP, and increasing amounts of AcrID1 before addition of double-stranded target DNA radio-labelled on the non-target strand (dsNTS). Upon incubation of the effector complex components with the dsNTS, but in the absence of AcrID1, several cleavage products appeared, which have been shown to be the result of the activity of the Cas10d HD domain (Fig. 4a)[20]. This activity was gradually reduced by addition of increasing amounts of recombinant AcrID1 and almost completely absent at a 1:4 molar ratio of Cas10d and the anti-CRISPR protein. Next, we investigated dsDNA binding by Cas10d/SS and the I-D backbone in the presence or absence of AcrID1 by electrophoretic mobility shift assay (EMSA, Fig. 4b). Pre-incubation of the main effector complex components (dsDNA, I-D backbone, Cas10d/SS, but not Cas3') with increasing amounts of AcrID1 significantly reduced the binding to the target (Fig. 4b, left) while pre-incubation of the effector complex components with the dsDNA target before the addition of AcrID1 did not show this effect (Fig. 4b, right panel).

To further delineate where AcrID1 interferes with effector complex assembly, we repeated the cleavage assay with preincubation of dsDNA with both backbone and Cas10d/SS or only one or the other, before addition of the remaining components, Cas3'/ATP and in the presence or absence of AcrID1 (Fig. 4c). This experiment confirmed that binding of both Cas10d/SS and backbone to dsDNA is both necessary and sufficient to prevent AcrID1 from interfering, while neither Cas10d/SS nor backbone binding to dsDNA is effective on their own. Taken together, this suggests that AcrID1, through its interaction with Cas10d, prevents formation of a cleavage-competent effector complex on DNA and moreover, that AcrID1 must interfere with Cas10d before a cleavage-competent complex with backbone and DNA has formed.

This result prompted the question of whether the extended conformation of Cas10d observed in the crystal structure is induced by AcrID1 binding or is an intrinsic feature of the protein. To answer this, we finally pursued a structure of the isolated Cas10d subunit. The isolated protein does not readily crystallise likely due to intrinsic flexibility, however, we were able to obtain crystals of a mildly trypsin-treated sample and determine the structure at 4.0 Å using molecular replacement (Supplementary Table 1). The new crystals contain two copies of Cas10d in the crystallographic asymmetric unit, for which the flexible N terminus (residues 1–13) that interacts with AcrID1, appears to have been cleaved off. Moreover, the Palm/RRM domain has been nicked by the protease resulting in the protein appearing as two bands on a denaturing polyacrylamide gel.

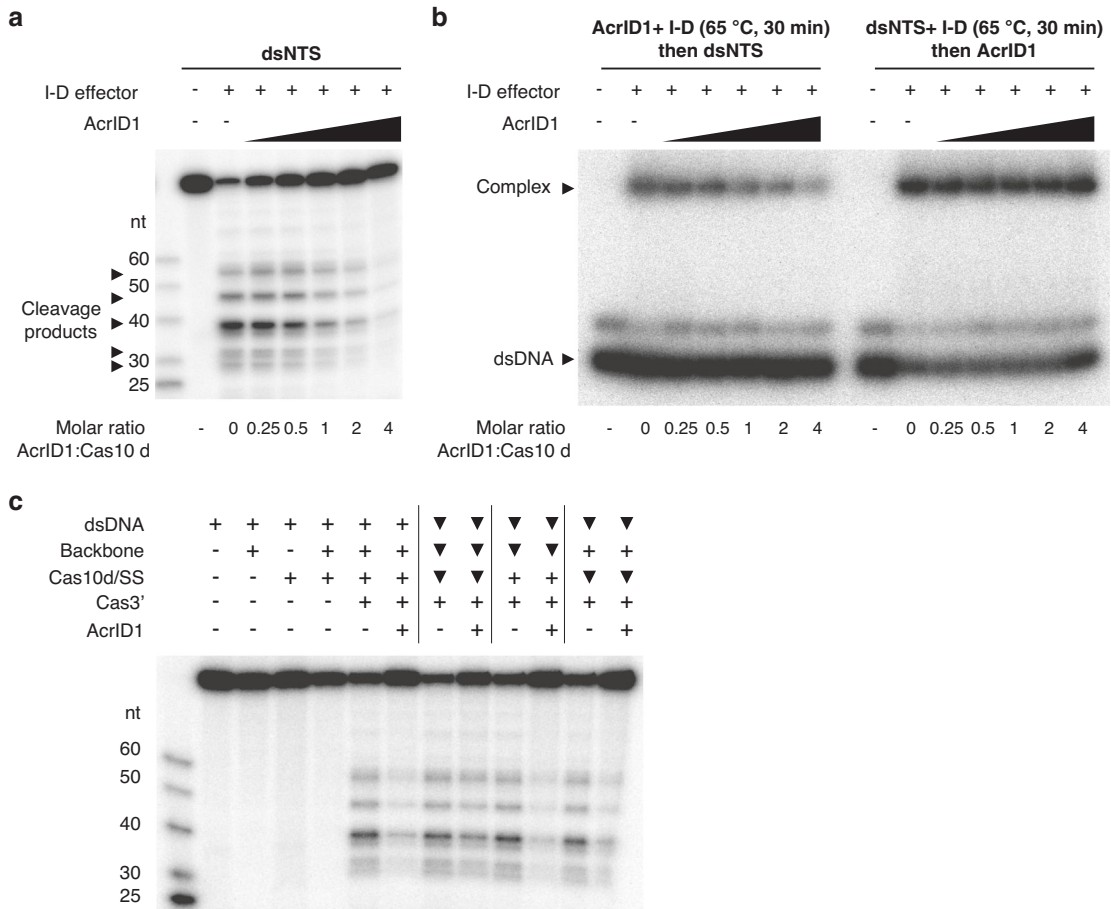

**Fig. 4 AcrID1 prevents formation of a cleavage-competent I-D effector complex. a** In vitro DNA cleavage assay. Each reaction contained the type I-D effector complex components (backbone, Cas10d/SS, and Cas3') and radio-labelled double-stranded target DNA (dsNTS), either in the absence or presence of increasing amounts of AcrID1, as indicated. The molar ratio of AcrID1 to Cas10d is indicated below the gel and major cleavage products are indicated with arrowheads on the side. The first lane is a molecular weight marker containing DNA fragments of known length as indicated (nt). **b** Electrophoretic mobility shift assay (EMSA) showing interaction between radio-labelled dsDNA (dsNTS) and the type I-D effector complex (backbone + Cas10d/SS) either in the absence or presence of increasing amounts of AcrID1, as indicated. Effector complex components (excluding Cas3') were pre-incubated for 30 min with either AcrID1 (left) or radio-labelled dsDNA (right). The positions of naked dsDNA and the DNA-bound effector complex ("complex") are indicated with arrowheads. **c** In vitro DNA cleavage assay with conditions as in (**a**). All reactions contained radio-labelled double-stranded target DNA (dsNTS) as well as the indicated type I-D effector complex components (backbone, Cas10d/SS, Cas3', and/or AcrID1). Components marked with a filled, black triangle were preincubated for 20 min prior to adding the other components. The first lane is a molecular weight marker containing DNA fragments of known length as indicated (nt). All gels are representative of three experiments with similar results. Source data are provided as a Source data file.

Nevertheless, the structure confirms the overall, extended conformation of Cas10d as observed in complex with AcrID1 (Fig. 5a). The helical insertion in the Palm/RRM domain, which interacts directly with AcrID1 in the complex structure, has shifted away from the core of the protein by about 5 Å in comparison to the complex with AcrID1 (Fig. 5a and Supplementary Fig. 7). Together, this suggests that the flexibility of Cas10d is mainly caused by the extended Palm/RRM domain. Consequently, it is likely that this domain gets organised when Cas10d joins the backbone complex and forms the complete I-D effector complex. In summary, we conclude that the extended structure of Cas10d is an intrinsic feature and not an effect of AcrID1 binding.

## Discussion

In this paper, we demonstrate that the *S. islandicus* type I-D large subunit, Cas10d, has a unique structure consisting of a type I-like

HD nuclease domain joined to a set of inactivated type III-like nucleotidyl transferase domains and a Cas11-like C-terminal domain. Moreover, we find that the overall domain organisation is unusual and more extended than other effector complex large subunits, likely due to an intrinsic flexibility of the Palm/RRM domain. AcrID1 is one of only three known SIRV anti-CRISPR proteins[23,33–35]. Acr proteins are typically small, highly divergent proteins that use unique features of their target Cas proteins for recognition[36]. Thus, sequence alignment of AcrID1 with Acr proteins targeting other type I CRISPR–Cas systems, such as *Pseudomonas aeruginosa* AcrF3 (anti-type I-F) or AcrE1 (anti-type I-E) reveal little similarity (Supplementary Fig. 8a). The interaction between AcrID1 and Cas10d appears uniquely tailored to the type I-D system, which mirrors what has been observed for other type I Cas-Acr complexes[29,37–39]. Even between closely homologous AcrID proteins from the *Sulfolobus islandicus* viruses SIRV2 and SIRV3, both targeting the type I-D CRISPR–Cas system, almost none of the interactions are

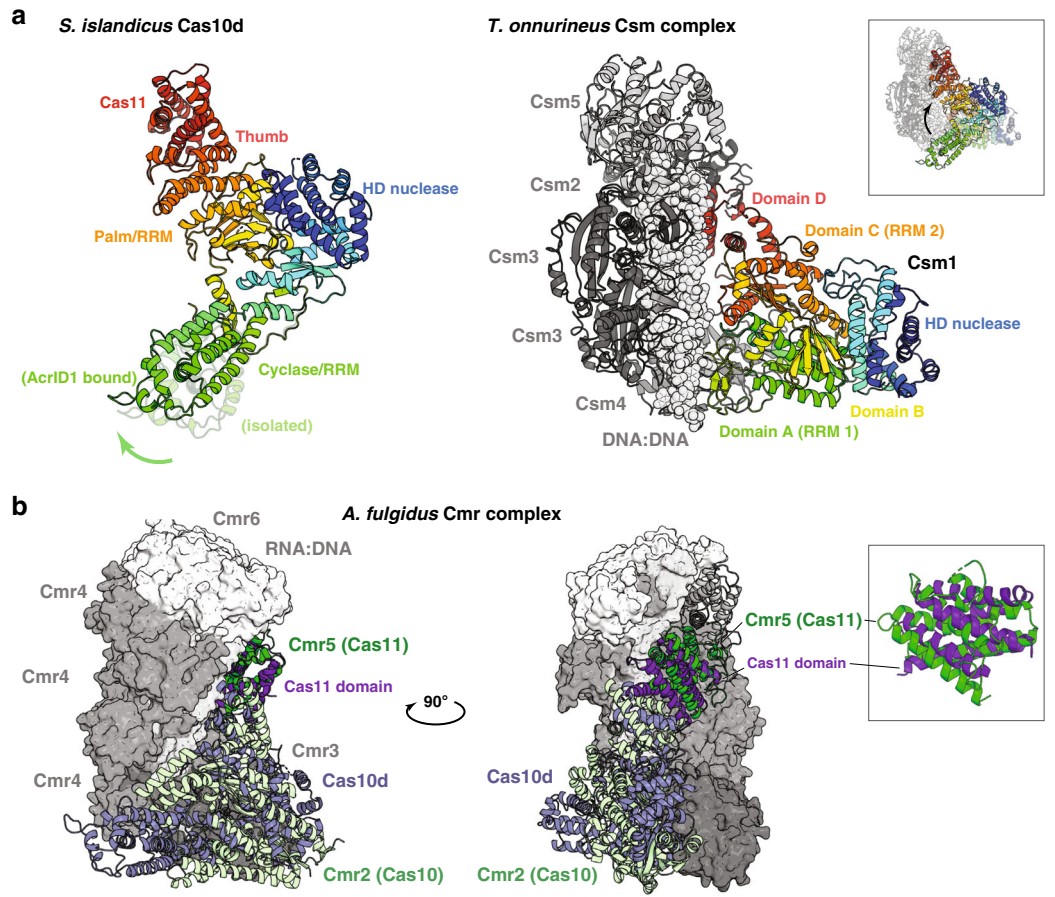

**Fig. 5 Comparison of Cas10d to type III effector complexes. a** Left, structure of *S. islandicus* Cas10d (this work) coloured in rainbow from blue (N terminus) to red (C terminus) with individual domains indicated and the conformation of the flexible Cyclase/RRM domain observed in the isolated structure of Cas10d shown in faint colours ("isolated"); Right, structure of the intact *T. onnurineus* Csm complex consisting of Csm1, Csm2, two copies of Csm3, Csm4, Csm5, and DNA as determined by cryo-EM (PDB ID 6MUT)[41]. The Csm1 subunit, which is homologous to Cas10d, is coloured using the same colour scheme. The inset shows an overlay of Cas10d and the Csm complex with an arrow indicating the putative shift of the Cyclase/RRM domain in Cas10d when bound to the backbone complex. **b** Structure of the *A. fulgidus* Cmr complex (PDB ID 3X1L) bound to a DNA:RNA hybrid substrate, consisting of Cmr2 (Cas10, light green) and Cmr5 (Cas11, dark green) in cartoon, and Cmr3 (Cas5), three copies of Cmr4 (Cas7), two copies of and Cmr6 (Cas7), and DNA/RNA shown as a grey surface. Superimposed on this, the structure of *S. islandicus* Cas10d (pale blue) with the Cas11 domain in purple[40]. The inset highlights the homology between Cmr5 and the Cas10d Cas11 domain (rmsd 7.5 Å over 610 atoms).

conserved suggesting that these proteins bind in very unique ways to their target proteins (Supplementary Fig. 8b).

The isolated and AcrID1-bound structures of Cas10d leave open the question of what the conformation of Cas10d is inside the type I-D effector complex. However, some information can be gleaned from comparison to the closest homologous type III effector complexes, such as the *T. onnurineus* Csm (type III-A) and *A. fulgidus* Cmr (type III-B) complexes determined by cryo-EM[40,41]. Comparison of Cas10d with the conformation of the corresponding large subunit in the Csm complex, Csm1, and taking the observed flexibility of Cas10d into account suggests that the helical extension of the Cyclase/RRM domain likely folds back up upon interaction with the effector complex and target DNA (Fig. 5a). Consequently, this also suggests that AcrID1, by stabilising the open conformation, sequesters Cas10d in a non-functional state preventing access to the effector complex (Figs. 5a, 6 and Supplementary Fig. 8a). This is consistent with our functional data and might also suggest that the AcrID1 and DNA binding sites on Cas10d overlap as has been seen for other Acr–Cas complexes[37,42]. On the other hand, once a cleavage-competent complex has been formed between Cas10d, backbone, and target DNA, AcrID1 is no longer able to interfere (Fig. 6). This principle of inhibition is likely unique and is supported by

recent findings showing that neither Cas10d nor the backbone complex can bind target DNA on their own[20]. It suggests that the interaction between Cas10d and target DNA is quite tenuous, which thus provides a window of opportunity for interference by an anti-CRISPR protein that can capture the large subunit. The additional interactions present in the circular 4:2 complex between Cas10d and AcrID1 as well as the stabilisation this complex confers to an intrinsically flexible subunit, likely play a role in allowing this to happen.

Finally, super-positioning of Cas10d on the large subunit of the *A. fulgidus* Cmr complex (Cmr2, type III-B) reveals a striking structural overlap of the Cas11-like domain of Cas10d with the Cas11 subunit of the type III-B system (Cmr5, rmsd 7.5 Å over 612 atoms), suggesting that the C-terminal domain of Cas10d in fact plays the role of the small subunit inside the type I-D effector complex (Fig. 5b). This is consistent with the lack of a *cas11* gene in the type I-D systems as well as previously observed structural similarities between small subunits and the C-terminal domain of several large subunits[43]. Moreover, recent studies have found that Cas10d co-purifies with a small 14 kDa protein (SS), which likely results from an alternative translation start position at the beginning of the Cas11 domain in the gene encoding the large subunit[20,44]. Comparison with the Cmr complex, which contains

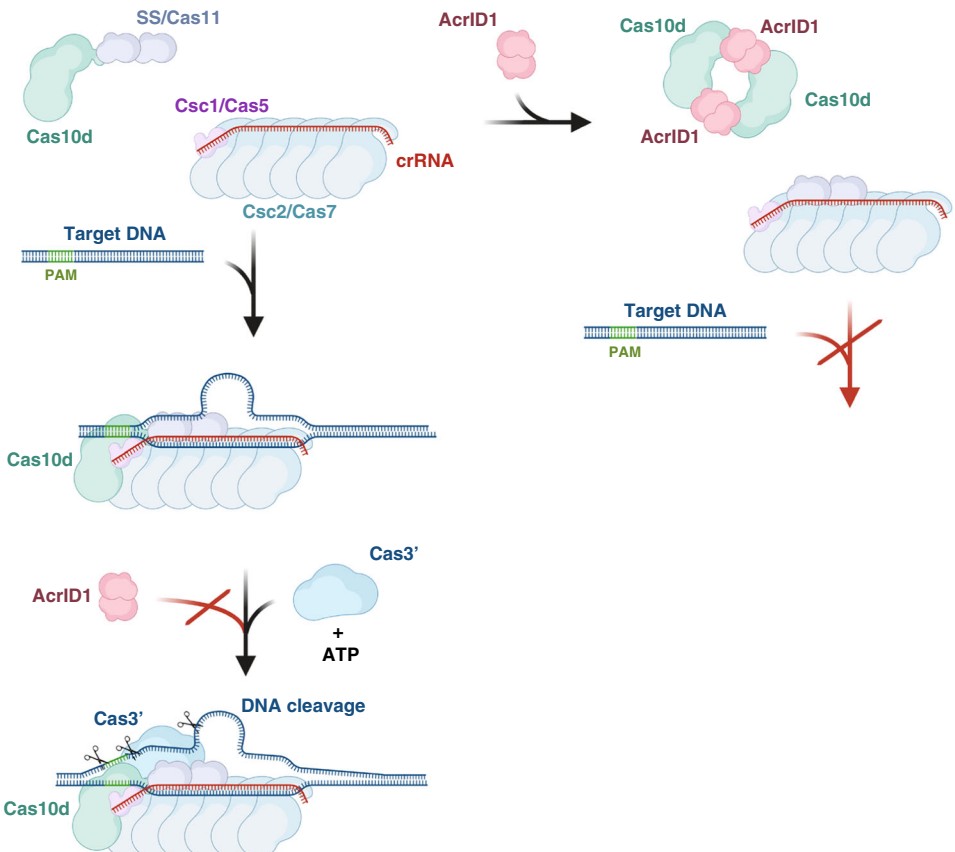

**Fig. 6 Model for inhibition of type I-D CRISPR–Cas systems by AcrID1.** Left, the backbone complex (consisting of Csc1/Cas5, Csc2/Cas7, and crRNA) can bind Cas10d/SS and target DNA, allowing for multiple cleavage events in the presence of Cas3' and ATP (lower left). The sequential binding order of Cas10d/SS, backbone, and Cas3' to target DNA is not known. AcrID1 interferes with Cas10d prior to DNA binding (top right), sequestering it in a non-functional, higher-order state and preventing interference. Once the backbone:Cas10d:DNA complex is formed, AcrID1 cannot interfere. Figure created with BioRender.com.

two copies of the Cmr5 (Cas11) small subunits further suggests that in the type I-D effector complex, these roles may be played by multiple copies of the C-terminal Cas11 domain of Cas10d generated by the alternative translation start. Structural studies of Cas10d in the context of the effector complex will hopefully shed light on this intriguing aspect of the type I-D effector complex[40,41].

## Methods

**Protein expression and purification.** The plasmids pET30a-cas10d and pET30a-acrID1 were used for expression of recombinant Cas10d and AcrID1, both with C-terminal His$_6$ tags[23]. Separate 2 L cultures of E. coli B834 (DE3) cells (Novagen) carrying plasmids expressing either Cas10d or AcrID1 were grown in lysogeny broth (LB) medium supplemented with 50 µg/mL kanamycin at 37 °C with shaking, until OD$_{600}$ reached approximately 0.6–0.8. The cells were harvested by centrifugation (1500 × $g$ for 10 min at 4 °C) and used to inoculate 6 L of M9 minimal medium (Molecular Dimensions), supplemented with 50 µg/mL kanamycin, glucose, vitamins and amino acids with the exception of L-methionine. The cells were grown for 3 h at 37 °C after which 20 mg/mL seleno–methionine (Se–Met, Molecular Dimensions) was added and the cells were further incubated until OD$_{600}$ reached 0.8. Protein expression was then induced with 0.5 mM isopropyl-β-D-thiogalactopyranoside (IPTG) and expression continued overnight at 20 °C. For protein purification, cells were resuspended in lysis buffer (50 mM Tris-Cl, pH = 8.0, 300 mM NaCl, 20 mM imidazole, 3 mM β-mercaptoethanol, β-ME) supplemented with 1 mM phenyl methyl sulfonyl fluoride (PMSF) and 10 µg/mL DNase I. The cells were lysed by sonication and cell debris removed by centrifugation (23,000 × $g$ for 45 min at 4 °C). The cell lysate was heated at 75 °C for 20 min to remove the bulk of E. coli proteins as a first step of purification, then centrifuged to clear the supernatant containing either Cas10d-His or AcrID1-His. The samples were then applied to a pre-packed 5 mL HisTrap column (Cytiva), pre-equilibrated in lysis buffer and eluted using a stepwise gradient into lysis buffer with increasing concentrations (30–300 mM) of imidazole. The eluate containing the protein of interest was then applied on a 1 mL Heparin column (Cytiva), pre-equilibrated in

50 mM Bicine, pH = 8.5, 50 mM NaCl, 5 mM β-ME, and eluted with a linear gradient into the same buffer containing 1 M NaCl. The sample was finally purified by size exclusion chromatography using a Superdex 200 increase 10/300 column (Cytiva), pre-equilibrated in 25 mM Hepes-NaOH, pH = 7.5, 150 mM NaCl, 5 mM β-ME. Peak fractions of Cas10d were pooled and concentrated to 5–15 mg/mL and used for crystallisation in isolation or to form the AcrID1-bound complex[23]. For complex formation, Cas10d and AcrID1 were mixed in a 1:2 (Cas10d:AcrID1) molar ratio and incubated at 75 °C for 1 min. The complex was then isolated using a Superdex 200 increase 10/300 column (Cytiva) pre-equilibrated in 30 mM Hepes-NaOH, pH = 7.5, 150 mM NaCl, 5 mM β-ME and concentrated to 4 mg/mL. For the isolated Cas10d structure, a sample of the protein was treated with bovine trypsin in a 1:500 (trypsin:Cas10d) molar ratio for 50 min at room temperature. The proteolysis reaction was quenched with 1 mM PMSF after which the sample was purified on a Superdex 200 increase 10/300 column (Cytiva), pre-equilibrated in 25 mM Na-Hepes, pH = 7.5, 300 mM NaCl, 5 mM β-ME. The protein eluted in a similar position to the uncut protein, but sodium dodecylsulphate polyacrylamide gel electrophoresis (SDS-PAGE) revealed the presence of two bands corresponding to approximately 45 and 50 kDa. The protein was concentrated to ~5 mg/ml using a 30 kDa MWCO VivaSpin sample concentrator (Sigma-Aldrich) before crystallisation screening. All protein concentrations were estimated using a NanoDrop measuring OD$_{280}$ and theoretical extinction coefficients calculated from the known protein sequences.

**Crystallisation and structure determination.** The purified Cas10d–AcrID1 complex was crystallised in 0.1 M Hepes, pH = 7.5, 4% (w/v) PEG 8000, 5% (v/v) ethylene glycol by sitting drop vapour diffusion prepared in Swissci MRC 2-drop crystallisation plates (Hampton Research) using a Mosquito® nanoliter protein crystallisation robot (TTP Labtech). Initial crystals appeared after two days at 19 °C and further optimised for data collection. All crystals were cryo-cooled in the reservoir buffer supplemented with 30% (v/v) ethylene glycol. After extensive crystal screening and optimisation, a single-wavelength Se anomalous dataset extending to 3.5 Å (CC$_{1/2}$ = 0.55 at 3.48 Å) was obtained at 100 K and 0.9793 Å X-ray wavelength, identified using a fluorescence energy scan for selenium using mxCUBE at the PETRA III beamline P14 in Hamburg, Germany (see

Supplementary Table 1 for crystallographic statistics). The raw data were processed and scaled using XDS and XSCALE[45], and data quality assessed using CCP4[46] and Phenix.xtriage[47]. Initial phases were calculated using Phenix.autosol and the map improved by density modification as implemented in the Phenix suite[47]. The initial Se-phased map was of sufficient quality to place an AcrID1 dimer based on the existing structure[23], while Cas10d was manually built in Coot[48] using the Se-phased experimental map and using the Se anomalous map as guide for methionine positions, which was critical to correct tracing. The model was improved by iterative model building and refinement in Phenix.refine. The clash score and Ramachandran statistics were improved using Namdinator[49], while shifts in registry and model geometry were corrected by model optimisation in ISOLDE[50] combined with refinement in Phenix.refine. The final structure has R-work=23.7% and R-free=26.6% and a relatively high, average B factor (137 Å²) likely owing to the high level of flexibility of Cas10d (Supplementary Table 1).

For the isolated Cas10d structure, initial crystals appeared by sitting drop vapour diffusion overnight at 19 °C in 100 mM Hepes-NaOH pH = 7.5, 8% (w/v) PEG 8000, 8 % (v/v) ethylene glycol and further optimised by addition of 1% (w/v) PEG 3350. The crystals were cryo-cooled in the reservoir buffer supplemented with 25% (v/v) ethylene glycol before native, monochromatic x-ray data extending to 4.0 Å were collected at the Diamond Light Source beamline I04 in Oxford, UK. The raw data were processed and scaled using XDS and XSCALE[45], and data quality was assessed using Phenix.xtriage[47]. The structure was determined using molecular replacement (Phenix.phaser)[51] using Cas10d from the Cas10d–AcrID1 complex as a search model. The crystals of isolated Cas10d contain two copies of the protein per crystallographic asymmetric unit, one of which appears to have a partial RRM/Cyclase domain, presumably due to the proteolytic treatment. In the other molecule, which is more intact, the RRM/Cyclase domain has shifted ~5 Å compared to the complex structure. Inspection of the packing further revealed that parts of the intact structures would clash in the new crystal, suggesting this is the reason that proteolysis allowed crystallisation of the isolated protein. Both chains were rebuilt in Coot and iteratively refined using Phenix.refine. The final structure has R-work = 34.3% and R-free = 37.0% and a very high, average B factor (228 Å²), presumably due to the additional flexibility induced by cleavage of the protein (Supplementary Table 1). All structure figures were created in PyMOL.

**Labelling of dsDNA target**. DNA oligonucleotides (Supplementary Table 2) were purchased from Integrated DNA Technologies (IDT) and purified by recovering the corresponding bands from a 12% denaturing polyacrylamide gel. The DNA of the non-target strand was 5'-labelled with [γ-³²P]-ATP (PerkinElmer) and T4 polynucleotide kinase (New England Biolabs) before double-stranded DNA (dsDNA) was generated by annealing the labelled non-target strand with 2-fold molar excess of unlabelled target strand.

**I-D CRISPR–Cas effector purification**. A *S. islandicus* LAL14/1 derivative strain lacking all CRISPR arrays[52] as well as the type I-A and type III-B systems but retaining the I-D effector module transformed with pEXA2-Csc1-S5–6 was used to purify backbone complex (Csc1, Csc2, and crRNA) essentially as described before[20]. Briefly, cells were grown in SCVY medium and the His-tagged complex purified using a His-trap column (Cytiva) followed by gel filtration using a Superdex 200 10/300 column (Cytiva) eluting in 20 mM Hepes-OH, pH = 7.5, and 300 mM NaCl. Recombinant Cas10d for the functional assays co-purified with small subunit (SS) and was purified from *E. coli* as described previously[23]. Recombinant MBP-Tev-Cas3'-His was overexpressed from pMAL-TEV-Cas3' in *E. coli* BL21 (DE3) Rosetta overnight at 20 °C and purified by HisTrap and gel filtration as described above, cleaved by Tev protease overnight at 4 °C, and purified by HisTrap and gel filtration again, eluting in a final buffer containing 20 mM Hepes-OH, pH = 7.5, and 300 mM NaCl.

**DNA cleavage assays**. DNA cleavage assays were conducted in 10 μL of reaction volume containing 0.02 mg/mL backbone complex, 0.03 mg/mL Cas10d, 0.01 mg/mL Cas3', 100 μM ATP, 20 nM dsDNA substrate and the indicated amounts of AcrID1 in a cleavage buffer (20 mM MES-NaOH pH = 6.0, 10 mM MgCl₂, 1 mM DTT) at 65 °C for 1 h. Reactions were stopped by adding 2× RNA loading dye (New England Biolabs). For electrophoresis, samples were heated for 3 min at 95 °C and analysed by 12% denaturing polyacrylamide gel electrophoresis (PAGE). For visualisation, a phospho-imager screen was exposed to the gel overnight and scanned with Typhoon FLA-7000. DNA cleavage gels (Fig. 4a, c) were run at identical conditions. Source data are provided as a Source data file.

**Electrophoretic mobility shift assays**. Each reaction mixture contained 0.02 mg/mL backbone complex, 0.03 mg/mL Cas10d, 20 nM dsDNA substrate and the indicated amounts of AcrID1 in cleavage buffer (20 mM MES-NaOH pH = 6.0, 10 mM MgCl₂, 1 mM DTT) and was incubated at 65 °C for 30 min. An equal volume of 2× native loading buffer (0.02% bromophenol blue, 40% (v/v) glycerol) was added and the samples were loaded on an 8% native PAGE running at 25 °C in 40 mM Tris, 20 mM acetic acid, pH = 8.4. Migration on the gel was visualised by exposure to a phospho-imager screen overnight, which was subsequently scanned on a Typhoon FLA-7000.

**Reporting summary**. Further information on research design is available in the Nature Research Reporting Summary linked to this article.

## Data availability

Coordinates and structure factors for Cas10d–AcrID1 (6THH) and for isolated Cas10d (6YES) have been deposited in the Protein Data Bank. Other publicly available datasets include PDB ID 6MUT, Cryo-EM structure of ternary Csm-crRNA-target RNA with anti-tag sequence complex in type III-A CRISPR–Cas system; PDB ID 3X1L, Crystal Structure of the CRISPR–Cas RNA Silencing Cmr Complex Bound to a Target Analog; PDB ID 4UW2, Crystal structure of Csm1 in *T.onnurineus*; PDB ID 4QQW, Crystal structure of *T. fusca* Cas3; PDB ID 4W8Y, Structure of full length Cmr2 from *Pyrococcus furiosus* (Manganese bound form); PDB ID 5B7I, Cas3-AcrF3 complex; PDB ID 3S4L, The CRISPR-associated Cas3 HD domain protein MJ0384 from *Methanocaldococcus jannaschii*; PDB ID 3SK9, Crystal structure of the *Thermus thermophilus* cas3 HD domain; and PDB ID 1JX4, Crystal Structure of a Y-family DNA Polymerase in a Ternary Complex with DNA Substrates and an Incoming Nucleotide. Source data are provided with this paper.

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

## Acknowledgements
We are thankful to Saravanan Panneerselvam and Thomas Lykke-Møller Sørensen for assistance during data collection at EMBL PETRA beamlines P14 and P13, respectively, and Kamel El Omari at Diamond Light Source for discussions on low-resolution model building. We are also thankful to Jesper Lykkegaard Karlsen for computational support and Tristan Croll for his help through using ISOLDE for improving the model. Funding for this work came from a Lundbeck Foundation post-doctoral grant to M.C.M. and grants from the Novo Nordisk Foundation to X.P. (grant no. NNF17OC0031154) and D.E.B. (grant no. NNF17OC0028072).

## Author contributions
D. E. B. and X. P. acquired funding and supervised the study. A. F. and X. P. made expression constructs while M.C.M. and L.B.V. expressed, purified, characterised and crystallised the Cas10d–AcrID1 complex and M.C.M. and D.E.B. carried out structure determination and refinement. L.B.V. purified and crystallised the isolated Cas10d protein and D.E.B. determined and refined the structure. A.F., J.L. and X.P. designed and carried out the functional studies. M.C.M., A.F., J.L., D.E.B. and X.P. prepared the figures and M.C.M., X. P., and D.E.B. wrote the paper.

## Competing interests
The authors declare no competing interests.
