## [Peer Review File · Nature Communications]

REVIEWER COMMENTS

Reviewer #1 (Remarks to the Author):

Cemre Manav et al describes a structural analysis of *Sulfolobus islandicus* Cas10d, both in isolation and in complex with the anti-CRISPR protein, AcrID1. They determined the structure of Cas10d:AcrID2 complex using X-ray crystallography with SeMet labeled protein. The structure is a 4:2 complex, consistent with SEC (Figure 1). Analysis of the domain organization of Cas10d, reveals is unique compared with type III Cas10 proteins. The HD domain of Cas10d is compared to the HD domains of Cas3 and type III Cas10 and is found to be a hybrid of the two. Additionally, the active site of the Cas10d HD domain is found to likely bind only one metal ion (Figure 2). Analysis of the Cyclase/RMM domain shows it lack features necessary for ATP binding and it therefore likely inactivated. The details of the interfaces between AcrID2 and Cas10d are described (Figure 3). To explore the consequences of AcrID1 binding the authors monitored the cleavage of a DNA target by the type I-D system with increasing concentrations of AcrD1. They found AcrD1 is a potent inhibitor of DNA cleavage. Further EMSA analysis revealed that this was likely due to preventing type I-D cascade from binding the target (Figure 4). Overall this is a well performed study and the conclusions are supported by the data. That said I have a few suggestions that I believe would strengthen the paper.

Suggestions:

1. It would be useful to show structural overlays of the Cas10d domains with those from other known cas10 structures. How similar are the various domains? What are the rmsd's in these alignments?
2. Biochemical data is presented that shows AcrID1 inhibits cleavage by the type I-D system by inhibiting DNA binding. A model is presented where AcrID1 does this by inhibiting binding of Cas10d to the type I-D backbone complex. However, there is very little direct data to support this other than the structural comparisons. SEC analysis of a mixture of AcrID2 and type I-D Cascade would directly support this hypothesis and should be readily achievable.
3. The ladder pattern of DNA cleavage (Figure 4A) is very reminiscent of type III cleavage of RNA. Could the authors comment on this?
4. The authors briefly describe a crystal structure of mildly-trypsin treated Cas10d, whose structure is consistent with the Cas10d:AcrID2 complex. This is a result and should be described in the results section.
5. For clarity, it might help to include topology diagrams as part of supplemental figure 3.
6. The buried surface area at the Cas10d:AcrID1 interface should be stated in the text.
7. Line 44: "Cascade" is an acronym that should be defined.
8. Line 131: "Cas3Tte" should this be Cas3Tth?

Reviewer #2 (Remarks to the Author):

Type I-D crisper systems have a Cas10d large subunit that represents a half-way-house between type I and III crisper systems. Previously, AcrID1 was shown to bind to Cas10d in dimeric form, forming a stable 2:2 complex (REF 21). Here, low resolution X-ray structures of Cas10d on its own (at 4 Å resolution) and in complex with AcrID1 (at 3.4 Å) are presented. The structure of the complex shows how the Acr traps the Cas10d subunit, forcing it to dimerise, in a way that could prevent the function

of the effector complex. The Acr cannot inhibit pre-formed 1-D complexes, which has implications for the assembly and activity of this effector.

The structure of isolated Cas10d is more extended and of lower resolution, perhaps reflecting intrinsic flexibility of the subunit in the absence of its partners. Here the low resolution could be problematic as it is at the lower limits of what can be determined by crystallography. There is a risk that the structure is biased by the use of the complex as a model to solve the apo protein structure. The validation report reveals lots of clashes and non-standard rotamers, which gives cause for concern. This uncertainty is compounded by lack of clear description of the methods used to refine the structures in the manuscript.

The strengths of the paper are as follows. Firstly, the first description of a Cas10d structure, which holds no surprises but nonetheless highlights the links to type III crisper systems. Secondly, the structural evidence for the interaction of AcrID1 with Cas10d and the hints that the Acr blocks assembly of a full effector complex.

The weaknesses of the paper include the aforementioned low resolution apo structure, the lack of a clear link between structure and mechanism, which is not fully developed in this paper, and the lack of adequate description of the methodology employed.

Specific points:

1. P4 line 58. Type III HD nuclease domains degrade ssDNA, but the evidence for co-transcriptional activity is very weak. Consider rephrasing.
2. P4 line 59. What does "natural nucleases" mean?
3. P11 line 226. There are three, not two, known SIRV Acrs – please add a reference for AcrIII-1.
4. In figure 4a, the cleavage products resemble the 6nt-spaced target RNA cleavage in type III systems catalysed by the Cas7 subunit. Can the authors comment on whether the observed nuclease activity is due to the HD domain of Cas10d or the backbone subunit? The model in figure 6 suggests that the only DNA cleavage observed is in the NTS and due to the action of Cas3' and the HD nuclease of Cas10d. The methods section suggests no ATP is present in this reaction, which would be required for the helicase activity of Cas3'. What does the pattern look like when ATP is present or Cas3' is absent?
5. Does pre-incubation of Cas10d with target DNA, but lacking the backbone subunits, still allow inhibition by AcrID1? This simple experiment would add more detail to the understanding of the mode of inhibition by the Acr. This is particularly relevant as Cas10d has been shown to recognise the PAM in target DNA by another group (<https://doi.org/10.1101/2020.03.14.991976>).
6. The methods state that key details including the purification of the backbone complex "will be published elsewhere" or are "in preparation". This is not really acceptable, so if the other paper is not publicly available in time then the details should be included here. Uploading a preprint to BioRxiv would be another option.
7. This mode of action of AcrID1, in blocking the assembly of an intact effector by sequestering one subunit, is unusual and indeed may be unique. In the discussion, could the authors expand on this and speculate on whether a loose interaction between the Cas10d and backbone of the complex is an intrinsic part of the mechanism of the effector?
8. The methods and software used to refine the structures must be added.

AUTHOR'S RESPONSE TO REVIEWER COMMENTS

Reviewer #1 (Remarks to the Author):

Cemre Manav et al describes a structural analysis of *Sulfolobus islandicus* Cas10d, both in isolation and in complex with the anti-CRISPR protein, AcrID1. They determined the structure of Cas10d:AcrID2 complex using X-ray crystallography with SeMet labeled protein. The structure is a 4:2 complex, consistent with SEC (Figure 1). Analysis of the domain organization of Cas10d, reveals is unique compared with type III Cas10 proteins. The HD domain of Cas10d is compared to the HD domains of Cas3 and type III Cas10 and is found to be a hybrid of the two. Additionally, the active site of the Cas10d HD domain is found to likely bind only one metal ion (Figure 2). Analysis of the Cyclase/RMM domain shows it lack features necessary for ATP binding and it therefore likely inactivated. The details of the interfaces between AcrID2 and Cas10d are described (Figure 3). To explore the consequences of AcrID1 binding the authors monitored the cleavage of a DNA target by the type I-D system with increasing concentrations of AcrD1. They found AcrD1 is a potent inhibitor of DNA cleavage. Further EMSA analysis revealed that this was likely due to preventing type I-D cascade from binding the target (Figure 4). Overall this is a well performed study and the conclusions are supported by the data.

Thank you very much!

That said I have a few suggestions that I believe would strengthen the paper.

Suggestions:

1. It would be useful to show structural overlays of the Cas10d domains with those from other known cas10 structures. How similar are the various domains? What are the rmsd's in these alignments?

We agree with the reviewer that this is an interesting point, however, the domains are too divergent for rmsd's to make sense. Supplementary Figure 2 provides an overview, coloured by domain and approximately aligned to show similar orientations, that gives an idea of the degree of (or lack of) similarity. A quick comparison of e.g. the dark blue Palm/RRM domain of Cas10d with the corresponding domain in the two other Cas10 structures (Csm1 and Cmr2) shows how different they are. The HD domain is the most conserved part of the structure, and in this case, we have included a thorough analysis and comparison to the closest, structural homologues, Cas3, in the paper.

2. Biochemical data is presented that shows AcrID1 inhibits cleavage by the type I-D system by inhibiting DNA binding. A model is presented where AcrID1 does this by inhibiting binding of Cas10d to the type I-D backbone complex. However, there is very little direct data to support this other than the structural comparisons. SEC analysis of a mixture of AcrID2 and type I-D Cascade would directly support this hypothesis and should be readily achievable.

The EMSA in Figure 4b shows that AcrID1 prevents the effector complex from binding to a dsDNA target, but we have no direct data on inhibition of Cas10d binding to the backbone.

The suggestion to check this in vitro by SEC is excellent, and we actually attempted this. The results, however, are not conclusive, because Cas10d and the backbone do not appear to form a complex stable enough to co-elute from SEC even in the absence of AcrID1. Nevertheless, the revised paper contains additional cleavage data that clearly delineate when AcrID1 must bind in order to prevent effector complex formation on DNA (Fig. 4c). These results strongly support the model presented in Fig. 6 and we believe that it also to some extent addresses the reviewer's question.

3. The ladder pattern of DNA cleavage (Figure 4A) is very reminiscent of type III cleavage of RNA. Could the authors comment on this?

This is correct and the details of dsDNA cleavage by the I-D effector complex are reported in the associated manuscript (Lin *et al.* "DNA targeting by subtype I-D CRISPR-Cas shows type I and type III features", DOI: gkaa749, recently accepted in *Nucleic Acids Research*) and also uploaded to the manuscript handling system along with this submission. The cleavage sites have been thoroughly analysed but do not fit with a ruler mechanism (See Lin *et al.* Fig. 1F-1J).

4. The authors briefly describe a crystal structure of mildly-trypsin treated Cas10d, whose structure is consistent with the Cas10d:AcrID2 complex. This is a result and should be described in the results section.

We agree with the reviewer on this point and have moved this part to the results section in the revised manuscript.

5. For clarity, it might help to include topology diagrams as part of supplemental figure 3.

We thank the reviewer for this excellent suggestion. Topology diagrams have been included in Supplementary Figure 3 of the revised manuscript as suggested, which also allow us to highlight the shared, structural features between the HD domains.

6. The buried surface area at the Cas10d:AcrID1 interface should be stated in the text.

The buried surface areas of the two different interfaces between Cas10d and the AcrID1 dimer have been included in the revised manuscript: "The buried surface areas between the AcrID1 dimer and the two molecules of Cas10d constitute approximately 930 and 880 Å² at the two distinct sites of interaction, and size exclusion chromatography supports that this higher-order complex represents the biologically relevant form.²³"

7. Line 44: "Cascade" is an acronym that should be defined.

Thanks. Definition included in the revised text.

8. Line 131: "Cas3Tte" should this be Cas3Tth?

We thank the reviewer for pointing this out. The residue numbers actually correctly refer to Cas3 from *Thermobaculum terrenum* (Tte), which we also analysed as part of the process.

The comparison to this structure was, however, excluded from the final manuscript as it is similar to Cas3 from *Thermobifida fusca* (Tfu) in this regard. The text has therefore been updated to include the correct residue numbers from Cas3 Tfu which is shown in both Figure 2b and with sequence in Supplementary Figure 4.

Reviewer #2 (Remarks to the Author):

Type I-D crisper systems have a Cas10d large subunit that represents a half-way-house between type I and III crisper systems. Previously, AcrID1 was shown to bind to Cas10d in dimeric form, forming a stable 2:2 complex (REF 21). Here, low resolution X-ray structures of Cas10d on its own (at 4 Å resolution) and in complex with AcrID1 (at 3.4 Å) are presented. The structure of the complex shows how the Acr traps the Cas10d subunit, forcing it to dimerise, in a way that could prevent the function of the effector complex. The Acr cannot inhibit pre-formed I-D complexes, which has implications for the assembly and activity of this effector.

The structure of isolated Cas10d is more extended and of lower resolution, perhaps reflecting intrinsic flexibility of the subunit in the absence of its partners. Here the low resolution could be problematic as it is at the lower limits of what can be determined by crystallography. There is a risk that the structure is biased by the use of the complex as a model to solve the apo protein structure. The validation report reveals lots of clashes and non-standard rotamers, which gives cause for concern. This uncertainty is compounded by lack of clear description of the methods used to refine the structures in the manuscript.

We agree with the reviewer that model bias can be an issue for molecular replacement carried out at low-to-modest resolution like in this case for the isolated Cas10d structure at 4.0 Å. However, following molecular replacement, the electron density map was very clear for the core structure and more importantly, showed that the Cyclase/RRM domain in one molecule had moved app. 5 Å compared to the search model, which to us suggested that the phases were strong enough to bring out the true structure. To demonstrate this to concerned readers, we have included a figure in the revision showing the MR solution (unmodified Cas10d structure from the complex) and the final isolated Cas10d model overlaid on the initial MR 2mFo-Fc density from Phenix.phaser (Supplementary Figure 7). We have also expanded the description of the crystallography methods in the revised version significantly.

The strengths of the paper are as follows. Firstly, the first description of a Cas10d structure, which holds no surprises but nonetheless highlights the links to type III crisper systems. Secondly, the structural evidence for the interaction of AcrID1 with Cas10d and the hints that the Acr blocks assembly of a full effector complex.

The weaknesses of the paper include the aforementioned low resolution apo structure, the lack of a clear link between structure and mechanism, which is not fully developed in this paper, and the lack of adequate description of the methodology employed.

Specific points:

1. P4 line 58. Type III HD nuclease domains degrade ssDNA, but the evidence for co-transcriptional activity is very weak. Consider rephrasing.

Thank you for this suggestion. We have removed this in the revised manuscript.

2. P4 line 59. What does “natural nucleases” mean?

We have rephrased this to "intrinsic nucleases" in the revised manuscript.

3. P11 line 226. There are three, not two, known SIRV Acrs – please add a reference for AcrIII-1.

Thanks for pointing this out. We have inserted a reference to the third type of SIRV Acr in the revised manuscript.

4. In figure 4a, the cleavage products resemble the 6nt-spaced target RNA cleavage in type III systems catalysed by the Cas7 subunit. Can the authors comment on whether the observed nuclease activity is due to the HD domain of Cas10d or the backbone subunit? The model in figure 6 suggests that the only DNA cleavage observed is in the NTS and due to the action of Cas3' and the HD nuclease of Cas10d. The methods section suggests no ATP is present in this reaction, which would be required for the helicase activity of Cas3'. What does the pattern look like when ATP is present or Cas3' is absent?

The cleavage pattern is described in more detail in the associated paper (Lin et al.) currently in press at Nucleic Acids Research and also included with this submission. The cleavages observed are due to the Cas10d HD domain, which is now pointed out in the revised manuscript. The associated paper also shows that Cas3' is needed for any cleavage to occur (Lin et al. Fig. 1E). In the absence of ATP in the reaction buffer (but presence of Cas3'), only the non-target strand is cleaved while both strands are cleaved when ATP is provided (Lin et al. Fig. 1F - 1J). ATP was also present in the cleavage experiments in Figure 4a of this paper but only NTS cleavage is visualised as this strand is labelled. We apologize that this information was missing in the submitted version, but it has been corrected in the revision.

5. Does pre-incubation of Cas10d with target DNA, but lacking the backbone subunits, still allow inhibition by AcrID1? This simple experiment would add more detail to the understanding of the mode of inhibition by the Acr. This is particularly relevant as Cas10d has been shown to recognise the PAM in target DNA by another group (<https://doi.org/10.1101/2020.03.14.991976>).

To address this question, we performed additional cleavage assays in which either Cas10d, backbone, or both, were pre-incubated with dsDNA prior to addition of AcrID1 and Cas3'/ATP (Fig. 4c). The results show quite conclusively that once a cleavage-competent complex of Cas10d/SS, backbone, and dsDNA is formed, AcrID1 can no longer interfere. However, neither interaction of Cas10d/SS nor backbone with dsDNA on their own is able to prevent the action of AcrID1. These additional pieces to the puzzle have allowed us to confirm the model in Fig. 6 more accurately, and we are thankful to the reviewer for suggesting this.

6. The methods state that key details including the purification of the backbone complex “will be published elsewhere” or are “in preparation”. This is not really acceptable, so if the other paper is not publicly available in time then the details should be included here. Uploading a preprint to BioRxiv would be another option.

The associated paper (Lin et al.) is now in press and has been cited more extensively in the revised manuscript. A draft of this paper was actually uploaded along with this submission but we apologize to the reviewers if this information was not made available to them at the time of reviewing the present manuscript. The final version of Lin et al. is attached to the revised manuscript submission.

7. This mode of action of AcrID1, in blocking the assembly of an intact effector by sequestering one subunit, is unusual and indeed may be unique. In the discussion, could the authors expand on this and speculate on whether a loose interaction between the Cas10d and backbone of the complex is an intrinsic part of the mechanism of the effector?

We thank the reviewer for raising this very interesting point. The tenuous character of the effector complex is supported both by our unpublished SEC data (see above) and the absence of Cas10d/Cas3' in the backbone purified from *Sulfolobus islandicus* (Lin et al. Fig. 1B). Together, this suggests that no or very little stable effector complex is present in cells in the absence of a DNA target. To accommodate the reviewer's point, we have revised the discussion of the revised manuscript.

8. The methods and software used to refine the structures must be added.

We apologize for this omission. The methods have now been updated to include both structures.

REVIEWERS' COMMENTS

Reviewer #1 (Remarks to the Author):

The reviewers were very responsive too the reviewers comments. I now recommend publication.

Reviewer #2 (Remarks to the Author):

The authors have addressed the points made by both reviewers in a constructive way and the revised paper is, in my opinion, now suitable for publication.

Malcolm White

AUTHOR'S RESPONSE TO REVIEWER COMMENTS

REVIEWERS' COMMENTS

Reviewer #1 (Remarks to the Author):

The reviewers were very responsive too the reviewers comments. I now recommend publication.

Thank you very much for your helpful review.

Reviewer #2 (Remarks to the Author):

The authors have addressed the points made by both reviewers in a constructive way and the revised paper is, in my opinion, now suitable for publication.

Malcolm White

Thank you very much for your openness and helpful review.